# Self, Semi and Fully Supervised Training for Autoencoders using Ternary Classification

## Abstract

Autoencoders are usually trained in a self-supervised fashion. In the context of anomaly detection, research shows that they can also be trained in a fully supervised one, using binary class labels, namely HEALTHY and FAULTY. However, when working with real world data, such an approach might not be suitable. It is hard to binary classify data coming from equipment that has been in operation for a long time, is affected by wear and tear. In additional, its real current health status is unknown. Moreover, historical data is not usually labeled, and only maintenance interventions are recorded. To alleviate this problem, a third label is introduced, UNKNOWN, which enables the autoencoder to learn the structure of healthy and faulty data from the correspondingly labelled data points. This structure is used in reconstructing the UNKNOWN inputs. This can increase the performance of autoencoders in a wide range of anomaly detection cases, especially when the timeseries data used to train the autoencoder comes from machines that have been in operation for a long time. This is especially relevant in the case of industrial machinery. Different label-aware loss functions which can enable the training of an autoencoder, using the three aforementioned labels, in any combination of self, semi and fully supervised training are investigated in this work. The loss functions presented in this paper enable an autoencoder to achieve particularly good anomaly detection performance on a clutch-slip detection dataset acquired from a test bench which simulates the drivetrain of an electric Range Rover Evoque. The dataset is presented in the appendix.

## 1 Introduction

Anomaly detection (AD) problems present a unique challenge: a dataset that is sparsely annotated. While autoencoders (AEs) work well for AD problems when trained in a self-supervised manner on only healthy data, in usual industrial applications there is no guarantee that the data supplied for the training of the AE is healthy – usually it comes from machines that have been in use for a long period of time, that have undergone maintenance for different problems and which are worn down (Kamat & Sugandhi, 2020; Pang et al., 2021).

Moreover, while AEs can work well even in this type of situations, the question arises if their performance can be improved by considering the labels of the data, i.e. by training them in a semi or even fully supervised fashion. This is especially important when considering corner cases (Bolte et al., 2019), where a specific behavior occurs hardly ever, but can have a significant impact on the functioning of a system. Machine learning (ML) algorithms can have difficulty detecting these cases, depending on the change in the behavior of the data. A usual approach to improve the performance of an anomaly detector is for an operator to label the data acquired during the corner case, and to fine-tune the ML algorithm on it.

In addition, if faulty data is present in the training set of the AE, the performance of the ML algorithm can decline. This happens because the AE is trained to reconstruct faulty data well - a behavior which is replicated in test and in production settings. The AE should be trained to correctly reconstruct healthy data and to badly reconstruct inputs coming from faulty equipment.

Considering all the above, in the context of AD, one should be be able to train and fine-tune an AE in any combination of the following learning paradigms: self, semi and fully supervised. Thus, the AE could use label information to improve its performance.

In this work it was investigated how can label information be integrated into an AE. There are several approaches that can be found in the literature:

- Modify the loss function of the AE to consider the possible binary labels of the data (Kanishima et al., 2022; Hanakata et al., 2020; Le et al., 2018);

- Add a classification head to perform binary classification using the latent space representation as its input (Gille et al., 2023; Gogna et al., 2017; Zhuang et al., 2015);

- Fine tune the autoencoder, already trained on unlabeled data, on a binary annotated subset (Meire et al., 2022);

- Use multiple AEs - either in a stack or in a Siamese fashion (Sun et al., 2022; Chen et al., 2023; Gao et al., 2015).

There is an important limitation in the current state of the art addressed in this work: the data used to train the autoencoder is considered healthy or faulty. However, in the real world, unlabeled data cannot be reliably labelled as such. It is highly likely that only a small subset of the data is healthy and faulty, while in the rest the behavior of the system falls somewhere in between. Thus, this large subset should be labelled as UNKNOWN. This approach allows the autoencoder to learn:

- To well reconstruct healthy data while learning the characteristics that define healthy data;

- To poorly reconstruct faulty data while learning the characteristics that define faulty data;

- To reconstruct unknown data somewhere in between, using the learnt characteristics that describe healthy and faulty data.

Different loss functions that enable an AE to be trained and/or later fined tuned in any combination of self, semi and fully supervised fashions are presented in this report. This approach was chosen because no further training of the AE is required, thus speeding up the anomaly detector development process.

The contributions of this work are twofold. Firstly, label-aware loss functions are proposed which can enable training and fine-tuning of AEs using any combination of self, semi or fully supervised learning. In contrast to the current state of the art, these loss functions use ternary labels and can take negative values. The second contribution is the usage of three different labels for the data, instead of only two.

The autoencoder used in this work and the baseline loss functions are shown in Section 2. The labels used for ternary classification, and the label-aware loss functions are presented in Section 3. A possible caveat from using the proposed loss functions and a mitigation procedure is shown in Section 4. The results are presented and discussed in Section 5. Conclusions and perspectives are shown in Section 6. The used dataset is presented in Appendix A.

## 2 BASELINE CASE FOR COMPARISON

The AE used in this work is shown in Figure 1.

Figure 1: The AE architecture used in this work.

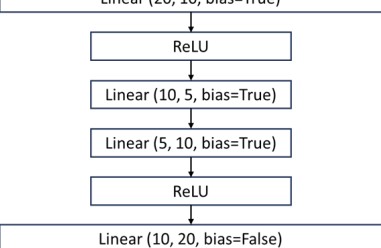

Table 1: The baseline loss functions.

| Name | Equation |
|------|----------|
| MSE | $L_{MSE} = \dfrac{1}{n} \sum_{i=1}^{n} (x_i - \hat{x}_i)^2$ |
| RMSE | $L_{RMSE} = \sqrt{\dfrac{1}{n} \sum_{i=1}^{n} (x_i - \hat{x}_i)^2}$ |
| L2-norm | $L_{l_2} = \sqrt{\sum_{i=1}^{n} (x_i - \hat{x}_i)^2}$ |
| Squared-L2 norm | $L_{l_2^2} = \sqrt{\sum_{i=1}^{n} (x_i - \hat{x}_i)^2}^{\,2}$ |

There are multiple loss functions that can be used to train autoencoders. The ones considered in this work are shown in Table 1.

where $n \in \mathbb{N}$ is the number of features.

These loss functions were chosen in this work because:

- Mean-Squared Error (MSE) is a standard loss for autoencoders;
- Root Mean Squared Error (RMSE) has less variation than MSE, thus the results should theoretically vary less in the different test settings;
- L2-norm is a standard distance and another frequently used loss function;
- Squared L2-norm is assumed to be more sensitive than MSE (as there is no division by the number of samples) and L2-norm (as there is no squared root being computed);

## 3 LABEL-AWARENESS

### 3.1 LABELS

To account for all categories of data that can be encountered in an AD system, the following labels are used:

- FAULTY = -1
- UNKNOWN = 0
- HEALTHY = 1

The HEALTHY and FAULTY labels are symmetric, with the UNKNOWN label in between. This allows the AD problem to be solved using an energy-based approach, more specifically contrastive learning (Khosla et al., 2020). A similarity can be drawn to the work of Liu & Gryllias (2020), but instead of putting the latent representations of healthy points inside a sphere and the representations of the others outside of the same sphere, the energy surface is designed, in a non-regular geometric way, such that the HEALTHY points reside in low-energy (i.e. reconstruction error) areas and the FAULTY ones in high-energy zones (Ranzato et al., 2007). As a consequence, the UNKNOWN ones will reside somewhere in between, their location being automatically determined by the AE according to what it learnt from the HEALTHY and FAULTY labeled data.

If a classification head were to be added to the AE, the value of the FAULTY labels can be easily changed to obtain a set of positive-valued labels – as it is required by loss functions used in the training of classifiers (e.g. cross-entropy loss).

The UNKNOWN label can be used to train an AE that performs identical to another one trained in a self-supervised fashion. Thus, all the modified loss functions presented in the following sub-section can be directly used with AEs that have already been trained and which only require fine-tuning.

## 3.2 LABEL-AWARE LOSS FUNCTIONS

The modified loss functions are said to be label-aware because they can consider labels in the training process, while maintaining full compatibility with self-supervised AE loss functions, through the usage of the UNKNOWN label. These label-aware loss functions are:

Table 2: The baseline loss functions.

| Name | Label-aware mechanism | Equation |
|---|---|---|
| MSE | exponents | $L_{la\_MSE} = \dfrac{1}{n} \sum_{i=1}^{n} (x_i - \hat{x}_i)^{2e_a}$ |
| | weights | $L_{la\_MSE} = \omega \dfrac{1}{n} \sum_{i=1}^{n} (x_i - \hat{x}_i)^2$ |
| RMSE | exponents | $L_{la\_RMSE} = \sqrt{\dfrac{1}{n} \sum_{i=1}^{n} (x_i - \hat{x}_i)^{2e_a}}$ |
| | weights | $L_{la\_RMSE} = \omega \sqrt{\dfrac{1}{n} \sum_{i=1}^{n} (x_i - \hat{x}_i)^2}$ |
| L2-norm | exponents | $L_{la\_l_2} = \sqrt{\sum_{i=1}^{n} (x_i - \hat{x}_i)^2}^{\,e_a}$ |
| | weights | $L_{la\_l_2} = \omega \sqrt{\sum_{i=1}^{n} (x_i - \hat{x}_i)^2}$ |
| Squared L2-norm | exponents | $L_{la\_l_2^2} = \sqrt{\sum_{i=1}^{n} (x_i - \hat{x}_i)^2}^{\,2e_a}$ |
| | exponents | $L_{la\_l_2^2} = \omega \sqrt{\sum_{i=1}^{n} (x_i - \hat{x}_i)^2}^{\,2}$ |

where $e_{la} = \begin{cases} -2, y_i = -1 \\ 1, y_i = 0 \\ 2, y_i = 1 \end{cases}$ is the label-aware exponent, $y_j \in \{-1, 0, 1\}$ is the label of the current train example, $\omega_j$ is the weight corresponding to the current example, $j \in 1, ..., N$ is the index of the current train example and $N \in \mathbb{N}$ is the number of examples in one batch. The exponent was placed in two distinct locations in different label-aware loss functions, to study the effect of its positioning on the obtained performance.

Different weights were investigated for these label-aware functions:

- Variants 1-5: These weights have the following general equation:

$$\omega_i = h y_i (1 + y_i) + u(1 - |y_i|) + f y_i (1 - y_i) \tag{1}$$

  where the $h, u, f \in \mathbb{R}$ are the weights associated with each label. Although these parameters can be optimized using an AutoML tool (He et al., 2021), several sets of values were considered in this work, to investigate their impact. These are shown in Table 3.

Table 3: The different variants of weights that were tested.

| Variant | Weight per label | | |
|---------|---------|---------|--------|
|         | Healthy | Unknown | Faulty |
| V1 | -10 | 1 | 10 |
| V2 | 1 | 0.001 | -1 |
| V3 | 10 | 0.001 | -10 |
| V4 | 5 | 1 | -50 |
| V5 | 5 | 1 | -500 |

- Variant 6: the class weights were computed by dividing the total number of labels by the number of labels which belong to each class.

$$\omega_{i_{v6}} = \{\frac{total\_num\_labels}{num\_healthy\_labels}, \frac{total\_num\_labels}{num\_unknown\_labels}, -\frac{total\_num\_labels}{num\_faulty\_labels}\} \quad (2)$$

## 4 NEGATIVE LOSS VALUES

FAULTY weights have negative values, so the value of the loss function can become negative. This is a desired property, as the typical training minimization objective will be turned into a maximization one. Thus, the AE will be rewarded for poorly reconstructing FAULTY inputs. However, care must be taken, as this can lead the optimization algorithm to explore regions of the training space where the loss is minimized to the expense of the good reconstruction of the HEALTHY inputs. An example of such undesired behavior is shown in Figure 4, where the AUROC is 52.240 and the EER is 95.519.

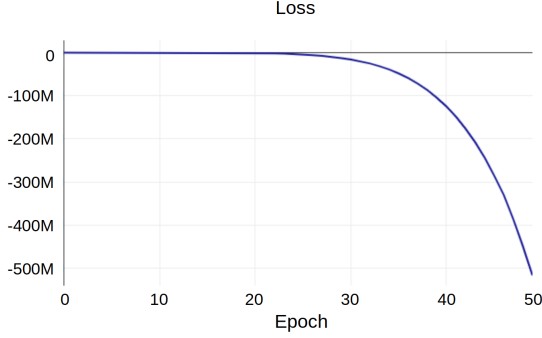

Figure 2: Negative loss values which lead to poor performance.

The solution is to limit the value of the loss, when it is negative. There are two possible approaches:

- A hard limit, such as a saturation;
- A soft limit, that reduces the absolute value without capping it explicitly.

The second approach was preferred in this work, because a hard cap might discard the knowledge gained from some of the negative values the loss function could take. The following strategies were tested to implement the chosen approach:

- Divide the negative loss values by a certain amount: a constant or the order of magnitude of the loss values;
- Compute the n-order root of the negative loss values. To calculate the n-order root of a negative number, it was decided to compute the n-order root of the absolute value of that number and then to multiply the result by -1.

The second method was chosen in this work, as it was easier to find a suitable n-order root that works well in all the tested situations. However, it should be noted that this is another hyperparameter that should be optimized – the results can change significantly based on its different values. This method can also be coupled with an early stopping mechanism. The study of different loss capping techniques should be the subject of further research.

The new modified loss functions are (only one is shown for brevity):

$$
L_{la\_l_2^2} = \begin{cases} \omega\sqrt{\sum_{i=1}^{N}(x_i - \hat{x}_i)^2}^2, & \omega\sqrt{\sum_{i=1}^{N}(x_i - \hat{x}_i)^2}^2 \geq 0 \\ \sqrt[\circ]{\omega\sqrt{\sum_{i=1}^{N}(x_i - \hat{x}_i)^2}^2}, & \omega\sqrt{\sum_{i=1}^{N}(x_i - \hat{x}_i)^2}^2 < 0 \end{cases}
\tag{3}
$$

The optimum value found for the n-order root is $o = 3.5$, which corresponds to computing the square of the number and then its seven order root.

By limiting the negative loss values, the new values of the performance metrics are AUROC = 99.996 and EER = 0.090. The new values of the loss function can be seen in Figure 4.

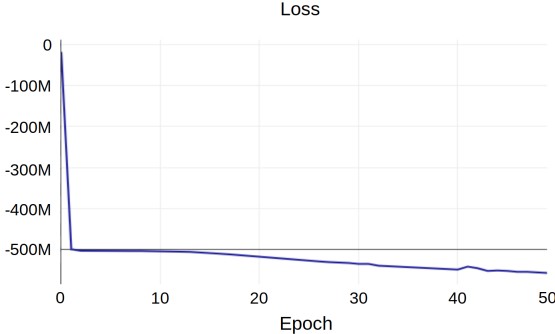

Figure 3: Soft-capped negative loss values which lead to good performance.

## 5  RESULTS AND DISCUSSION

The proposed label-aware loss functions were tested on a dataset constructed by Flanders Make which is available either for use in joint projects or for purchase. The data was acquired from a test setup which replicates the rear differential unit of an electric Range Rover Evoque (Make, 2020). The dataset used in this work is comprised of eleven files, shown in Table 4. More details about the dataset and the test setup can be found in Appendix A.

Table 4: Files used in the experiments.

| File ID | 1 | 2 | 3 | 4 | 5 | 6 | 7 | 8 | 9 | 10 | 11 |
|---|---|---|---|---|---|---|---|---|---|---|---|
| Labels | H | HU | HU | H | HUF | H | HU | HU | HUF | HU | HUF |
| No. clutch slips | 0 | 0 | 0 | 0 | 5 | 0 | 0 | 0 | 1 | 0 | 1 |

The experiments were conducted according to the test matrix shown in Table 5.

The results of the experiments are shown in Tables 6 and 7. In all experiments the ADAM optimizer was used with a learning rate of 0.003. The autoencoder was trained using a batch size of 8, and it was tested using a batch size of 1. The best results for each experiment are highlighted in bold. For testing purposes, binary classification is performed and UNKNOWN labeled data points are considered to be HEALTHY, to not trigger false alarms. The binary classification is based on the MSE between the input and the output of the AE.

Table 5: The test matrix

| ID | IDs of files in the train set, grouped by the contained labels | | | ID of test file | Motivation |
|---|---|---|---|---|---|
| | H | HU | HUF | | |
| 1 | 1, 4, 6 | N/A | N/A | 5 | Ideal train set for an AE, using only healthy data. |
| 2 | 1, 4, 6 | 2, 3, 7, 8, 10 | N/A | 5 | Extended train set, no faulty data included. |
| 3 | 1, 4, 6 | N/A | 9 | 5 | Investigation of the effects of a medium size fault in the train set. |
| 4 | 1, 4, 6 | 2, 3, 7, 8, 10 | 9 | 5 | Investigation of the effects of a medium size fault in the extended train set. |
| 5 | 1, 4, 6 | N/A | 11 | 5 | Investigation of the effects of a large size fault in the train set. |
| 6 | 1, 4, 6 | 2, 3, 7, 8, 10 | 11 | 5 | Investigation of the effects of a large size fault in the extended train set. |
| 7 | 1, 4, 6 | N/A | 9, 11 | 5 | Investigation of the effects of medium and large size faults in the train set. |
| 8 | 1, 4, 6 | 2, 3, 7, 8, 10 | 9, 11 | 5 | Investigation of the effects of medium and large size faults in the extended train set. |

Details regarding the size of the clutch slips used in the train set can be found in Appendix B.

The performance of the autoencoder trained using the baseline loss function, MSE, increases between experiments 1 and 2 due to the added data. However its performance decreases when faults and more UNKNOWN labeled data points are introduced in the train set. The performance also decreases monotonically with the amplitude of the faults present in the train set.

In experiment 1, where no UNKNOWN or FAULTY data points are present in the train set, the autoencoders trained using label-aware loss functions consistently outperform the one trained using MSE loss. While the different in AUROC is rather small, the improvement in EER approaches 30%. In experiment 2, where UNKNOWN labeled data is introduced, the performance gain from using label-aware loss functions decreases, and some cases it slightly degrades. This happens because the label-aware loss functions force the autoencoder to learn from the structure of the data. However, because of the test methodology, the autoencoder it is expected to well reconstruct UNKNOWN labeled data points.

In experiments 3 to 8, it can be seen that the exponents-based loss does not usually perform well when faults are present in the train set. It seems that simply inverting the loss function does not provide enough of a reward for the poor reconstruction of FAULTY labeled data points. The only exception is when using L2 norm loss. This may be due to the combination of the lower values that the L2 norm has before exponentiation, in contrast to its squared counterpart, and to the position of the exponent, in contrast to RMSE loss. Thus, the location of the label-aware exponent does play a role. A thorough investigation into the effect of its position and its values should also be the subject of future work.

On the issue of weights-based label-awareness, it can be seen that the greater the amplitude of the fault(s) present in the training set, the better the AEs perform. A similar conclusion can be drawn regarding the number of faults present in the dataset. It should be noted that the parameters of the weights-based label-aware loss functions should be optimized, to maximize the performance. Although intuition would tempt one to assign a large absolute value to the weight of the "FAULTY" labels, the experiments prove that it does not necessarily lead to better performance. This may happen because the search space becomes dotted with very steep peaks with a small "footprint" which are either overshot by the optimizer, or which act as a trap for it. Additionally, assigning a small weight to UNKNOWN labels, thus trying to ignore them, also seems to be a good strategy when the expected outcome is a binary classification.

Another noticeable aspect is that the weights computed automatically, using the number of class labels and the total amount of labels present in the train set, are not necesarily the optimal ones.

Table 6: Detailed results for experiments 1 to 4

| | | Exp. ID: | 1 | | 2 | | 3 | | 4 | |
|---|---|---|---|---|---|---|---|---|---|---|
| | | Metrics: | AUROC | EER | AUROC | EER | AUROC | EER | AUROC | EER |
| Baseline | | MSE | 97.09 | 12.28 | 97.90 | 8.51 | 95.62 | 14.88 | 96.85 | 12.46 |
| Label-aware loss functions | Exponents | RMSE | **97.93** | 10.22 | 95.35 | 12.46 | 49.92 | 58.33 | 39.41 | 54.66 |
| | | MSE | 97.06 | 9.05 | 95.35 | 12.46 | 46.32 | 58.33 | 39.41 | 54.66 |
| | | $L_2$ | 97.30 | 10.93 | 97.52 | 9.23 | 87.94 | 23.03 | 97.42 | 10.93 |
| | | $L_2^2$ | 96.60 | 12.46 | 97.28 | 11.56 | 67.42 | 45.79 | 85.75 | 26.34 |
| | Weights v1 | RMSE | 97.65 | 9.05 | **98.56** | **8.33** | 97.54 | 9.32 | 97.53 | 12.46 |
| | | MSE | 97.49 | 9.50 | 97.93 | 8.42 | 95.77 | 8.33 | 99.98 | 0.27 |
| | | $L_2$ | 97.41 | 10.04 | 97.39 | 9.95 | 97.60 | 9.68 | 97.76 | 10.66 |
| | | $L_2^2$ | 97.49 | 9.50 | 97.93 | 8.42 | 95.95 | 12.37 | 98.78 | 4.12 |
| | Weights v2 | RMSE | 97.67 | **8.96** | 98.50 | 8.33 | 96.79 | 10.31 | 99.88 | 1.97 |
| | | MSE | 97.49 | 9.50 | 97.91 | 8.51 | 95.82 | 12.46 | 99.77 | 4.12 |
| | | $L_2$ | 97.30 | 10.31 | 97.45 | 9.77 | 97.59 | 9.59 | 97.55 | 10.39 |
| | | $L_2^2$ | 97.49 | 9.50 | 97.87 | 8.51 | 87.24 | 22.22 | 99.94 | 1.25 |
| | Weights v3 | RMSE | 97.65 | 9.05 | 98.51 | **8.33** | 97.51 | 10.84 | 99.57 | 4.12 |
| | | MSE | 97.49 | 9.50 | 97.88 | 8.51 | 73.33 | 73.33 | 99,.98 | 0.45 |
| | | $L_2$ | 97.42 | 9.68 | 97.35 | 10.03 | 97.56 | 9.50 | 97.54 | 10.39 |
| | | $L_2^2$ | 97.49 | 9.50 | 97.84 | 8.60 | 99.44 | 4.12 | 98.89 | 4.12 |
| | Weights v4 | RMSE | 97.65 | 9.05 | **98.56** | **8.33** | 99.27 | 5.56 | 92.68 | 12.46 |
| | | MSE | 97.49 | 9.50 | 97.93 | 8.42 | 98.49 | 8.33 | **100.00** | **0.00** |
| | | $L_2$ | 97.41 | 10.04 | 97.39 | 9.95 | **99.77** | **3.76** | 99.96 | 1.08 |
| | | $L_2^2$ | 97.49 | 9.50 | 97.93 | 8.42 | 99.34 | 4.12 | 99.76 | 4.12 |
| | Weights v5 | RMSE | 97.65 | 9.68 | **98.56** | **8.33** | 97.09 | 4.12 | 99.99 | 0.18 |
| | | MSE | 97.49 | 9.50 | 97.93 | 8.42 | 95.06 | 8.33 | 99.97 | 0.63 |
| | | $L_2$ | 97.41 | 10.04 | 97.39 | 9.95 | 97.39 | 8.33 | **100.00** | **0.00** |
| | | $L_2^2$ | 97.49 | 9.50 | 97.93 | 8.42 | 98.18 | 8.33 | 99.66 | 4.12 |
| | Weights v6 | RMSE | N/A | N/A | N/A | N/A | 96.29 | 12.46 | 99.91 | 0.18 |
| | | MSE | N/A | N/A | N/A | N/A | 94.16 | 8.33 | **100.00** | **0.00** |
| | | $L_2$ | N/A | N/A | N/A | N/A | 97.32 | 10.57 | 92.56 | 16.67 |
| | | $L_2^2$ | N/A | N/A | N/A | N/A | 95.95 | 12.46 | 99.56 | 4.12 |
| Improvement over baseline [%] | | | +0.59 | +35.64 | +0.67 | +2.16 | +4.33 | +295.2 | +3.25 | +inf |
| Improvement over best baseline [%] | | | -0.23 | -6.30 | +0.67 | +2.16 | +1.91 | +126.2 | +2.14 | +inf |

This is another motivation to optimize the weights of the label-aware loss functions, possibly using an AutoML tool.

## 6 Conclusions and Perspectives

Label-aware loss functions were introduced in this paper, which enable the training and fine-tuning of AEs in any combination of self, semi and fully supervised learning using ternary classification. The proposed label-aware loss functions are designed to help the AE learn the structure of the healthy and faulty parts of the dataset using the corresponding labels, and to use this information in reconstructing UNKNOWN inputs.

The loss functions can have negative values, which reward the AE for poor reconstructions of FAULTY inputs. However, the behavior of the AE might diverge from what is expected, and it might reconstruct all possible inputs poorly. Thus, the negative values of the loss functions should be capped. A simple capping mechanism based on computing the n-order root of the negative loss

Table 7: Detailed results for experiments 5 to 8

| | | Exp. ID: | 5 | | 6 | | 7 | | 8 | |
|---|---|---|---|---|---|---|---|---|---|---|
| | | Metrics: | AUROC | EER | AUROC | EER | AUROC | EER | AUROC | EER |
| Baseline | | MSE | 95.37 | 14.07 | 96.30 | 12.19 | 95.17 | 12.46 | 94.75 | 12.99 |
| Label-aware loss functions | Exponents | RMSE | 47.24 | 48.57 | 52.11 | 41.67 | 51.42 | 58.33 | 50.78 | 50.00 |
| | | MSE | 49.26 | 57.89 | 52.11 | 41.67 | 51.36 | 53.76 | 50.78 | 50.00 |
| | | $L_2$ | 98.56 | 8.33 | 98.80 | 4.12 | 89.72 | 20.88 | 95.38 | 12.46 |
| | | $L_2^2$ | 92.68 | 19.18 | 90.11 | 14.43 | 78.88 | 28.58 | 73.08 | 33.33 |
| | Weights v1 | RMSE | **100.00** | **0.00** | 99.99 | 0.09 | **100.00** | **0.00** | 99.99 | 0.18 |
| | | MSE | **100.00** | **0.00** | 95.72 | 8.33 | **100.00** | **0.00** | 99.99 | 0.18 |
| | | $L_2$ | 99.26 | 4.12 | 98.80 | 4.12 | 99.18 | 5.29 | 98.95 | 4.12 |
| | | $L_2^2$ | **100.00** | **0.00** | 99.99 | 0.09 | **100.00** | **0.00** | **100.00** | **0.00** |
| | Weights v2 | RMSE | **100.00** | **0.00** | 99.99 | 0.27 | 99.87 | 0.27 | **100.00** | **0.00** |
| | | MSE | **100.00** | **0.00** | **100.00** | **0.00** | 99.99 | 0.18 | **100.00** | **0.00** |
| | | $L_2$ | 99.99 | 0.09 | 99.94 | 0.18 | 99.20 | 4.12 | 99.20 | 7.35 |
| | | $L_2^2$ | **100.00** | **0.00** | **100.00** | **0.00** | **100.00** | **0.00** | **100.00** | **0.00** |
| | Weights v3 | RMSE | **100.00** | **0.00** | **100.00** | **0.00** | 99.87 | 0.27 | 99.99 | 0.18 |
| | | MSE | 99.99 | 0.09 | 99.92 | 1.88 | **100.00** | **0.00** | **100.00** | **0.00** |
| | | $L_2$ | 98.97 | 4.12 | 97.12 | 4.12 | 99.72 | 2.69 | 99.04 | 4.12 |
| | | $L_2^2$ | 99.99 | 0.18 | 99.99 | 0.27 | **100.00** | **0.00** | **100.00** | **0.00** |
| | Weights v4 | RMSE | 97.31 | 5.38 | 96.91 | 6.18 | 85.04 | 29.93 | 82.39 | 35.22 |
| | | MSE | **100.00** | **0.00** | 95.14 | 12.46 | **100.00** | **0.00** | **100.00** | **0.00** |
| | | $L_2$ | 99.37 | 1.25 | 99.86 | 0.27 | 99.60 | 0.81 | 99.82 | 0.36 |
| | | $L_2^2$ | 99.92 | 1.97 | **100.00** | **0.00** | **100.00** | **0.00** | **100.00** | **0.00** |
| | Weights v5 | RMSE | 89.70 | 20.61 | 98.21 | 3.58 | 71.06 | 57.89 | 97.81 | 4.39 |
| | | MSE | **100.00** | **0.00** | 97.98 | 8.33 | **100.00** | **0.00** | 99.99 | 0.18 |
| | | $L_2$ | 82.62 | 34.77 | 95.79 | 8.42 | 85.08 | 29.84 | 67.97 | 64.07 |
| | | $L_2^2$ | 98.09 | 4.12 | 99.99 | 0.27 | 99.99 | 0.18 | 99.95 | 1.17 |
| | Weights v6 | RMSE | 84.50 | 31.00 | 68.50 | 62.99 | 68.77 | 62.46 | 88.89 | 22.22 |
| | | MSE | 99.89 | 1.79 | 92.04 | 12.55 | **100.00** | **0.00** | 91.91 | 12.55 |
| | | $L_2$ | 99.81 | 4.12 | 99.71 | 1.25 | **100.00** | **0.00** | 99.99 | 0.18 |
| | | $L_2^2$ | **100.00** | **0.00** | 99.90 | 2.24 | **100.00** | **0.00** | **100.00** | **0.00** |
| Improvement over baseline [%] | | | +4.85 | +inf | +3.84 | +inf | +5.07 | +inf | +5.53 | +inf |
| Improvement over best baseline [%] | | | +2.14 | +inf | +2.14 | +inf | +2.14 | +inf | +2.14 | +inf |

was presented in this paper. This mechanism is shown to lead to very good results, with the downside of introducing another hyperparameter to be optimized.

The first perspective for future research is to investigate other capping mechanisms - a promising lead is to compute the logarithm of the absolute value of the negative loss. A second perspective is the investigation of the effect of the different positions of the label-aware exponent on the performance of the resulting AEs. The third perspective entails testing the proposed methodology on other architectures of artificial neural networks (e.g. variational AE) and on other datasets.

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

## A  DATASET

The proposed label-aware loss functions were tested on a dataset constructed by Flanders Make which is available either for use in joint projects or for purchase. The dataset was acquired from the Multi-Load Drivetrain Test Cell (MLDTC) of Flanders Make. This experimental setup replicates the rear differential unit of an electric Range Rover Evoque, with the goal of understanding how different contexts can influence the functioning of the vehicle.

A diagram of the setup is shown in Figure 4. The MLTDC consists of a main motor (i.e. the output of a gearbox) which drives the propshaft connected to the rear differential unit (RDU). The differential with a pump and two clutches are controlled individually by an open-source embedded device and are mounted in the original subframe of the vehicle. There are two load motors (i.e. wheels) connected to the outputs of the clutches.

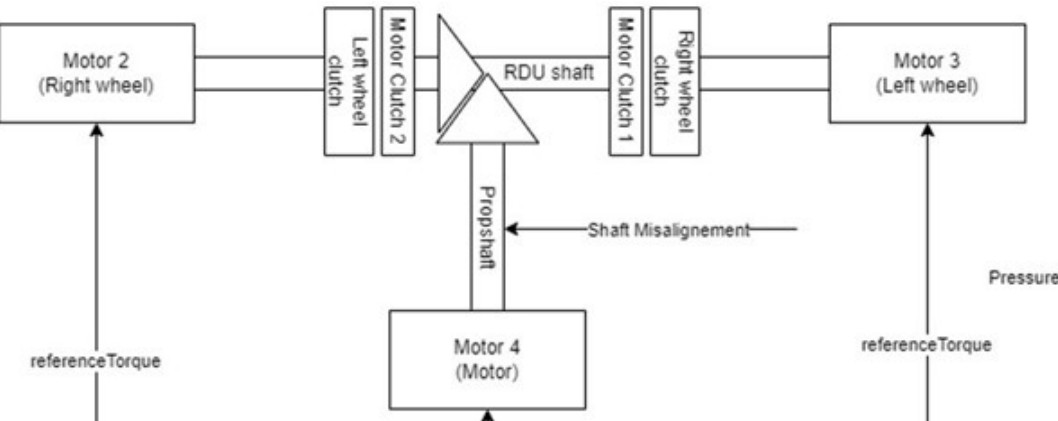

Figure 4: Schematic of the MLTDC.

The contexts which can be simulated are:

- Driver behavior;
- Vehicle type:
  - Mass;
  - Drag area;
  - Tire friction coefficient;
  - Tire radius;

    – Maximum acceleration;
- Clutch type.

Different anomalies can be introduced in the setup, to study their effect on the system:

- Clutch slip;
- Shaft misalignment;
- Sudden flat tire;
- Bearings failure;
- Blocked bearing.

The test setup is controlled using the reference signals shown in Table 8.

Table 8: MLDTC reference signals.

| Signal | Unit of Measure |
|---|---|
| propshaft reference speed | rpm |
| left clutch reference pressure | Pa |
| right clutch reference pressure | Pa |
| vehicle reference velocity | mps |

In the data used for this work, the reference signals for the platform were computed by a Simulink model of the Range Rover Evoque. The input of the model was the vehicle reference velocity (the same one fed to the MLDTC) and its outputs were the other reference signals. The vehicle reference velocity was acquired from driving a Range Rover Evoque in Sweden. The actual car was not fitted with all the sensors present on the test platform, so the MLTDC setup was used to gain deeper insights into its behavior.

Measurements were acquired from the platform at a sampling rate of 1 kHz and they are listed in Table 9.

Table 9: MLDTC measured signals.

| Reference | Unit of Measure | Comment |
|---|---|---|
| propshaft speed | rpm | |
| left clutch pressure | Pa | |
| left wheel speed | rpm | |
| left wheel torque sensor raw | Nm | Raw measurements from the torque sensor. |
| left wheel torque sensor | Nm | Filtered measurements from the torque sensor. |
| propshaft torque sensor raw | Nm | Raw measurements from the torque sensor. |
| propshaft torque sensor | Nm | Filtered measurements from the torque sensor. |
| right clutch pressure | Pa | |
| right wheel speed | rpm | |
| right wheel torque sensor raw | Nm | Raw measurements from the torque sensor. |
| right wheel torque sensor | Nm | Filtered measurements from the torque sensor. |
| vehicle simulated velocity | mps | The velocity attained by the Simulink model. |
| vehicle velocity | mps | |

The acquired data was stored in .mat files. It was later copied into HDF5 files (the standard data storage format in the "Monitoring" group of Flanders Make). The data was also labelled, using the following procedure:

- If the difference in speed between the propshaft and the wheel shafts is lower than a certain value, then the data point was labelled as HEALTHY;

Table 10: Labels distribution per split.

| | Labels | | |
|---|---|---|---|
| Split | Healthy | Unknown | Faulty |
| train | 10078 | 110 | 20 |
| test | 1116 | 24 | 24 |

- If the difference in speed between the propshaft and the wheel shafts was higher than a certain value, then the data point was labelled as FAULTY;
- Otherwise the data was labelled as UNKNOWN.

The data was also down-sampled from 1kHz to 10 Hz, to reduce the file sizes.

A significatly larger dataset was acquired from the MLTDC platform. Only a small subset was used in this work. Two splits were created from the dataset: a train and a test one. Concerning the dataset used for experiment 8, the distribution of the labels is shown in 10.

Although there are more instance of clutch slip present in the test file (file ID 5), their length is shorter than that of the ones present in the train set. It was thus investigated whether shorter duration anomalies can also be detected.

The goal of this work was clutch slip detection. The signals considered in this work were "propshaft speed" and "left wheel speed". Taking into account the used signals, the problem can be solved using simpler signal processing algorithms – thus the utilization of an AE is not a contribution of this work.

## B    CLUTCH SLIP VALUES

The clutch slip values are shown in Figure 5.

Figure 5: Fault sizes: in red is the clutch slip amplitude in file 9, and in blue the amplitude of the fault in File 11.

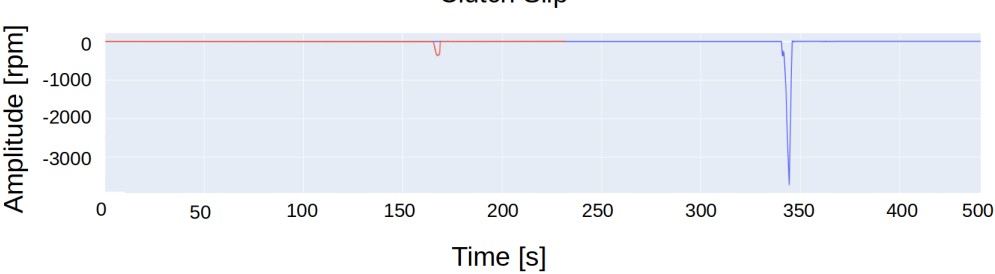

