# OpenReview forum: "Self, Semi and Fully Supervised Training for Autoencoders using Ternary Classification"
_ICLR.cc/2024/Conference — ICLR 2024 Conference Withdrawn Submission_

### Official Review · Reviewer_GBH8 · 2023-10-28

**Soundness:** 1 poor
**Presentation:** 1 poor
**Contribution:** 1 poor
**Rating:** 1
**Confidence:** 5

**Summary:**

This paper studies a third label (UNKNOWN) that enables the autoencoders to learn the structure of healthy and faculty data from the correspondingly labeled data points.

**Strengths:**

NA.

**Weaknesses:**

1. The technical novelty is very limited. And the proposed method does not make sense to me. It is hard to read the paper.
2. I did not get the whole point of the motivation and practical meaning of the paper. Perhaps it needs a better presentation.
3. They only evaluated on one dataset, which is not convincing enough. Also it is hard to interpret the table.
4. Figure 4 should be of higher resolution.

**Questions:**

See weaknesses.

---

> ### Author Response · Authors · 2023-11-14
>
> Dear Reviewer,
>
> We thank you for the time and effort you allocated to check our submission. We also thank you for pointing out how we may improve our work. We will take into account your comments while we will improve the presentation of our results to better show them on a more application-oriented dissemination channel, which should be more appropriate for our work.
>
> The paper was written by practitioners in anomaly detection and it may be better targeted towards other similar professionals. Unfortunately the limited number of pages prevented us from clearly presenting the whole context to people of different professional backgrounds.
>
> We agree that more thorough testing on other datasets is required to cross-validate our results and we thank you for your suggestion. We shall also take care to update the resolution of the problematic figure.
>
> We have decided that it is better to withdraw our paper, as it is not appropriate for ICLR.

---

### Official Review · Reviewer_EDWo · 2023-10-31

**Soundness:** 1 poor
**Presentation:** 2 fair
**Contribution:** 1 poor
**Rating:** 3
**Confidence:** 4

**Summary:**

The authors propose labeled-aware loss functions for autoencoders.
Based on 4 baseline loss functions, MSE, RMSE, L2-norm and
Squared-L2-norm, they introduce additional parameters depending on the
labels (healthy, faulty, and unknown).  One approach is to include the
parameter as an exponent (exponent-based), another is to multiple by
the parameter (weight-based).  To prevent negative loss values, they
use a soft limit by reducing the absolute values.

Using a dataset that contains parts that are healthy, faulty, and
unknown condition, they set up different scenarios for supervised,
semi-supervised, and unsupervised learning.  Label-aware loss
functions seem to perform better.

**Strengths:**

The proposed idea is relatively straight forward.  A real-world dataset
from automobile parts is used.

**Weaknesses:**

The 4 proposed baseline loss functions seem to be very similar in
terms of minimization.  Particularly, MSE and Squared-L2 norm differ
by a constant factor, which can be absorbed by the learning rate.
Autoencoders can be used for learning features without labels.
However, the authors propose modifying the autoencoder loss functions
with labels, which makes learning the autoencoder supervised.  Then a
comparison with regular supervised learning without an autoencoder
would be important.

Details are in questions below.

**Questions:**

1.  Table 1: Squared-L2 norm: the square and square root can be
cancelled out.  Squared-L2 norm / n is equivalent to MSE, so the two
loss functions differ by a constant (n is the same in both loss
functions).  That is, when Squared-L2 norm is minimized, MSE is
minimized.  When MSE is minimized, RMSE is also minimized.  Similarly,
when Squared-L2 norm is mininized, L2-norm is minimized.  Hence, the
four loss functions seem to be equivalent in terms of minimization.
Any insights on why these four loss functions are chosen?


2.  Table 2: $e_la$ and $e_a$ are different, but only $e_la$ is
discussed in the text.  What are the values for $e_a$?


3.  Equation 1: what is the motivation?


4.  Table 4: What are H, HU, HUF?  Does each file have multiple
instances?

5.  Tables 6 and 7: how do you get inf % improvement?

---

> ### Author Response · Authors · 2023-11-14
>
> Dear Reviewer,
>
> We thank you for the time and effort you allocated to check our submission and for your in-depth report. We also thank you for pointing out how we may improve our work.
>
> Concerning your questions:
> 1. We chose these loss functions because they are commonly used to train autoencoders. Although they share the same minima, they do not lead to the same outcome because their gradients differ. Therefore, the weights of the resulting AEs are different because of the finite number of training epochs (even though it may be argued that they can asymptotically lead to the same weights values). The same holds for the similarity you correctly observed between the Squared L2-norm and the MSE. In addition, the location of the fault-aware exponent is different between the Squared L2-norm and the MSE, which leads to different behavior.
> 2. That is a typographical error on our part. Thank you for having noticed it!
> 3. In equation 1, it is shown that each label is associated with a different weight. We agree that it is not sufficiently clear and we thank you for having brought that to our attention!
> 4. H, HU and HUF represent the labels used to annotate the data in each file. "H" stands for "HEALTHY", "HU" for "HEALTHY and UNKNOWN" and "HUF" for "HEALTHY, UNKNOWN and FAULTY". Thus, a file which contains the labels "HU" contains data points which are labeled "HEALTHY" or "UNKNOWN".
> 5. We calculate the improvement as a percentage. In percentage terms, any number is infinitely greater than 0. In this case it may be better to compute the improvement using absolute values, to avoid any confusion.
>
> We have decided that it is better to withdraw our paper, as it is not appropriate for ICLR.

---

### Official Review · Reviewer_wAkT · 2023-11-01

**Soundness:** 2 fair
**Presentation:** 2 fair
**Contribution:** 2 fair
**Rating:** 3
**Confidence:** 4

**Summary:**

This paper studies using autoencoders to perform anomaly detection, and the central observation is that labeled data with unknown type can improve model performance.

**Strengths:**

The paper made certain observations that have practical values.

**Weaknesses:**

The technical novelty is insufficient for ICLR.  The techniques proposed in the paper seem to be standard.

**Questions:**

NA

---

> ### Author Response · Authors · 2023-11-14
>
> Dear Reviewer,
>
> We thank you for the time and effort you allocated to check our submission. This work would indeed be better suited for a more application-oriented dissemination channel.
>
> We have decided that it is better to withdraw our paper, as it is not appropriate for ICLR.